# Characterization and Electrical Properties of PVA Films with Self-Assembled Chitosan-AuNPs/SWCNT-COOH Nanostructures

**DOI:** 10.3390/ma13184138

**Published:** 2020-09-17

**Authors:** Israel Ceja, Karla Josefina González-Íñiguez, Alejandra Carreón-Álvarez, Gabriel Landazuri, Arturo Barrera, José Eduardo Casillas, Víctor Vladimir A. Fernández-Escamilla, Jacobo Aguilar

**Affiliations:** 1Departamento de Física, Centro Universitario de Ciencias Exactas e Ingeniería, Universidad de Guadalajara, Blvd. M. García Barragán # 1421, C.P. 44430 Guadalajara, Mexico; iscean12@yahoo.com.mx; 2Departamento de Química, Centro Universitario de Ciencias Exactas e Ingeniería, Universidad de Guadalajara, Blvd. M. García Barragán # 1421, C.P. 44430 Guadalajara, Mexico; karlajgi@hotmail.com; 3Departamento de Ciencias Naturales y Exactas, Centro Universitario de los Valles, Universidad de Guadalajara, Carretera Guadalajara-Ameca Km. 45.5, C.P. 46600 Ameca, Mexico; ale_carreon_a@yahoo.com.mx; 4Departamento de Ingeniería Química, Centro Universitario de Ciencias Exactas e Ingeniería, Universidad de Guadalajara, Blvd. M. García Barragán # 1421, C.P. 44430 Guadalajara, Mexico; gabriel.landazuri@academicos.udg.mx; 5Departamento de Ciencias Básicas, Centro Universitario de la Ciénega, Universidad de Guadalajara, Avenida Universidad No. 1115, C.P. 47810 Ocotlán, Mexico; arturobr2003@yahoo.com.mx; 6Departamento de Ciencias Tecnológicas, Centro Universitario de la Ciénega, Universidad de Guadalajara, Avenida Universidad No. 1115, C.P. 47810 Ocotlán, Mexico; duart_casillas@hotmail.com (J.E.C.); vladkrm@hotmail.com (V.V.A.F.-E.)

**Keywords:** gold nanoparticles, self-assembly, nanostructures, films, electrical properties

## Abstract

Nanostructured films with electrical conductivity in the semiconductor region were prepared in a polymeric matrix of poly(vinyl alcohol) (PVA) with nanostructures of chitosan-gold nanoparticles (AuNPs)/single-wall carbon nanotubes carboxylic acid functionalized (SWCNT-COOH) (chitosan-AuNPs/SWCNT-COOH) self-assembled. Dispersion light scattering (DLS) was used to determine the average particle sizes of chitosan-AuNPs, *z*-average particle size (*Dz*) and number average particle size (*Dn*), and the formation of crystalline domains of AuNPs was demonstrated by X-ray diffraction (XRD) patterns and observed by means of transmission electron microscopy (TEM). The electrostatic interaction was verified by Fourier transform infrared spectroscopy (FTIR). The electrical conductivity of PVA/chitosan-AuNPs/SWCNT-COOH was determined by the four-point technique and photocurrent. The calculated *Dn* values of the chitosan-AuNPs decreased as the concentration of gold (III) chloride trihydrate (HAuCl_4_·3H_2_O) increased: the concentrations of 0.4 and 1.3 mM were 209 and 90 nm, respectively. Average crystal size (*L*) and number average size (*D*) of the AuNPs were calculated in the range of 13 to 24 nm. Electrical conductivity of PVA/chitosan-AuNPs/SWCNT-COOH films was 3.7 × 10^−5^ σ/cm determined by the four-point technique and 6.5 × 10^−4^ σ/cm by photocurrent for the SWCNT-COOH concentration of 0.5 wt.% and HAuCl_4_·3H_2_O concentration of 0.4 mM. In this investigation, the protonation of the amine group of chitosan is fundamental to prepare PVA films with nanostructures of self-assembled chitosan-AuNPs/SWCNT-COOH.

## 1. Introduction

Biodegradable polymers have a broad spectrum of applications. These polymers, with good mechanical properties and biocompatibility, can be used in biomedical implants, and the structure of these polymers is very important due to their sensitivity to hydrolytic degradation [1]. The incorporation of gold nanoparticles into these polymers leads to an increase in their electrical conductivity [2]. Chitosan is a biodegradable polymer that possesses the ability to absorb heavy metals or form chelate metal ions [3], and to stabilize and reduce gold nanoparticles [4,5] by functional groups (-NH_2_ and -OH). Nanostructured materials such as chitosan-Au nanoparticles have good compatibility and a high surface area, which may have catalytic activity [6,7] and can be used as sensors [8]. Zhang et al. [9] prepared an electrochemical immunosensor composed of gold nanoparticles (AuNPs) and chitosan films with adhesive capacity, where chitosan avoided the agglomeration of the AuNPs. The polymers play an important role because they maintain the stability of nanoparticles, have good physical properties and chemical stability and are low cost [10]. Moreover, aliphatic polymers such as PVA are widely used in different technology areas because they have good chemical and thermal properties and inherent non-toxicity. PVA/glucose oxidase (GOD) nanofibrous membranes were prepared by electrospinning to immobilize enzymes [11], and poly(ethylene glycol)-modified glucose oxidase was immobilized in PVA to be employed as a biosensor [12,13]. Gold nanoparticles inside polymer films are often used in biosensors because they allow a better electron transfer between the enzyme and the electrode in the electrochemical process [14,15]. Parlak et al. [16] prepared AuNPs–structured MoS_2_ nanosheets for electrochemical glucose biosensors, where the nanoparticles increased the electron transfer, creating a higher electroactive surface area, and had a more conductive interlayer for electron transfer. Furthermore, carbon nanotubes (CNT) have optic and electrical properties for technological applications, biomedical images and transistors [11]. In addition, they have quite an ample surface area, higher chemical stability, high mechanical and thermal properties and great capacities of adsorption and electric conductivity [8,17,18,19]. AuNPs have been deposited into CNT to form nanostructured compounds, which can be used as electrodes, catalysts and sensors [20]. Nanocomposites based on decorative multi-walled carbon nanotubes (MWCNT)-AuNPs encapsulated in a polymeric matrix improve the stability, reproducibility and sensitivity [21]. Functionalized carbon nanotubes improve the chemical–physical interaction with other molecules to yield nanostructures with well-defined shapes, such as self-assembled nanostructures. Therefore, the potential to build nanoscale architectures is necessary to understand the interactions between nanoparticles and molecules with CNT, which can be interactions of Van der Waals, charge transfer and covalent bonds [22]. The formation of self-assembled nanostructures is very important in optics, electronics and sensors applications, and their properties depend on the shape and size of the nanostructures [23].

We developed nanostructured films by using PVA/chitosan-AuNPs/(SWCNT-COOH), which possess electrical properties in the semiconduction region. In this study, we report a simple route to prepare films with embedded self-assembled nanostructures, which was carried out by the synthesis of AuNPs from chitosan, formation of self-assembled chitosan-AuNPs/(SWCNT-COOH) nanostructures and finally films formation. The sizes of chitosan-AuNPs were controlled keeping an accurate measurement of the concentration of HAuCl_4_·3H_2_O and chitosan; the length and structure were important to reach this goal because their low molecular weight allowed them to contract to yield structures that were quasi-spherical, and protonated amine groups allowed the electrostatic interaction with SWCNT-COOH. The process is quite new, inasmuch as it is not necessary to carry out chemical reactions for the formation of self-assembled structures, resulting in good conductivity, as is the case with similar systems, where it is necessary to use several reagents to stabilize the self-assembled structures [22,24].

## 2. Materials and Methods

### 2.1. Reagents

PVA (Golden Bell^MR^ Reagents, high-viscosity and water-solvable, Materiales y Abastos Especializados, S.A. de C.V. Jalisco, Mexico), deacetylated chitin, poly(D-glucosamine) (low-molecular weight chitosan, Sigma-Aldrich, Buchs, Switzerland), SWCNT-COOH (DxL 4–5 nm × 0.5–1.5 μm of Sigma-Aldrich, Steinheim, Germany), Gold(III) chloride trihydrate (HAuCl_4_·3H_2_O, Sigma-Aldrich, purity ≥ 99%, Stenheim, Germany), glacial acetic acid (CH_3_CO_2_H, Golden Bell^MR^ Reagents, Materiales y Abastos Especializados, S.A. de C.V. Jalisco, Mexico, ACS reagent, 99.7%), glycerol (GC, Sigma-Aldrich, ACS reagent 99.5%), sulfuric acid (H_2_SO_4_, J. T. Baker, Fisher Scientific SL, Madrid, Spain, ACS reagent) and bidistilled and deionized water were used as received.

### 2.2. Synthesis of Gold Nanoparticles and Chitosan-AuNPS/SWCNT-COOH Nanostructures

AuNPs were synthesized using chitosan as a reductant and stabilizer agent. For the synthesis, a CH_3_CO_2_H aqueous solution at 1 wt.% and different solutions (0.4, 0.7, 1.0 and 1.3 mM) of HAuCl_4_·3H_2_O were prepared. Afterwards, a solution was prepared with 0.5 wt.% of chitosan (weight percentage based on the mass solution added of HAuCl_4_·3H_2_O) dissolved in 5 mL of the CH_3_CO_2_H aqueous solution at 1 wt.%. Finally, 4 g of each solution of HAuCl_4_·3H_2_O was mixed in the chitosan solutions. The mixture was sonicated for 120 min in a Branson 3510 sonicator and stored for 24 h to complete the formation of AuNPs (chitosan-AuNPs dispersions). This procedure was applied to each HAuCl_4_·3H_2_O concentration. The synthesized chitosan-AuNPs dispersions were collocated again under sonication and were added to the SWCNT at different concentrations (0.1, 0.3 and 0.5 wt.%; the percentage was based on the mass solution added of HAuCl_4_·3H_2_O); immediately, 0.05 mL of sulfuric acid at a concentration of 0.1 N was added, and mixed at 60 °C for 60 min.

### 2.3. Nanostructured Film Formation of PVA/Chitosan-AuNPs/(SWCNT-COOH)

The films were formed in two steps. In the first step, a polymeric solution of PVA was used to prepare the film, with a ratio of 1/20 PVA/H_2_O (wt./wt.) by continuous stirring at 400 rpm and at 90 °C until complete dissolution of PVA. Then, 1.2 g of the PVA solution was added into a square glass container measuring 2 × 2 cm, with walls 0.5 cm high, and GC was added as the plasticizing agent (5 wt.% based on the mass of PVA) and mixed by sonication for 10 min. Finally, the mixture was placed in a Thermo Scientific^TM^ (Ohio, OH, USA) vacuum-heating oven at 50 °C for two days to form films between 0.25 and 0.3 mm in thickness. For the second step, the PVA film inside the glass container was immersed in a Branson Ultrasonic Corp. ultrasonic bath at 60 °C, taking care for it not to be in contact with the water. Right away, we added the dispersion of the nanostructures of chitosan-AuNPs/SWCNT into a glass container; the liquid of the dispersion formed a “mud”. This allowed for the nanostructures of chitosan-AuNPs/SWCNTs to distribute into the polymeric matrix, and the sonication continued until the solvents were evaporated and the film was formed.

### 2.4. Characterization

Distribution particle sizes of the chitosan-AuNPs were measured via dynamic light scattering (DLS), with a Zetasizer ZS90 (Malvern instruments, Malvern, UK) at 25 °C. *Dn, Dz* and the polidispersity index (*PDI*) were calculated from distributions of chitosan-AuNPs particle size by the equations shown in Appendix A. AuNPs were characterized by (TEM, JEOL, Tokyo, Japan) in a JEOL 1010 equipment in order to determine the structure and calculate the average particle size (*D*). The AuNPs were separated from chitosan by ultracentrifugation at 7000 rpm in a centrifuge (VWR scientific model V, California, CA, USA) and dispersed with bidistilled water by sonication. A drop was redispersed onto a carbon-coated copper grid and afterwards observed through TEM. The PVA/chitosan-AuNPs/(SWCNT-COOH) nanostructured films were frozen-dried in liquid nitrogen and then fractured to determine the size and morphology in a TESCAN MIRA 3LMU scanning electron microscope (SEM, TESCAN, Brünn, Czech Republic). The formation of AuNPs was confirmed by UV–Vis spectroscopy (Cary-300) (Varian, Palo Alto, CA, USA) with a sample holder for liquids, and the spectra were recorded in the wavelength range between 400 and 800 nm. The UV–Vis spectra obtained from an UV-Vis-NIR UV-3600 spectrophotometer (Shimadzu, Tokyo, Japan) were used to determine the band gap energy (*Eg*) of the PVA/chitosan-AuNPs/(SWCNT-COOH) nanostructured films, with a films sample holder, and the spectra were recorded at 190–800 nm. From the spectra, the *Eg* values were calculated by using the Kubelka–Munk equation (1) according to direct transition, whereas: n = 1/2 [25,26].
(1)(F(R)hv)1n=B(hv−Eg)
where *F(R)*, *h*, *v*, B, *Eg* and *n* are the Kubelka–Munk function, Planck constant, light frequency, specific constant of material, energy band gap and nature of the transition, respectively. The band gaps were obtained by plotting (*F(R)hv*)^2^ of the Kubelka–Munk function versus *hv* and extrapolating the straight line of the curve to zero in the *hv* axis, this relation being known as a Tauc graph [27]. The structure of gold nanoparticles was analyzed by XRD in a STOE & CIE GMBH (Darmstadt, Germany) apparatus equipped with a copper source at 30 kv and 15 mA, with Kα radiation of 1.5406 A°, and an X-ray irradiated the power samples between 5 and 80 range over scattering angle 2θ.

An FTIR Perkin Elmer Spectrum One spectrometer (Shelton, CT, USA) in a spectral ranging from 400 to 4000 cm^−1^ was used to identify the physical interaction between chitosan and SWCNT-COOH. The chitosan, SWCNT-COOH and chitosan-AuNPs/SWCNT-COOH were collocated into KBr tablets obtained by compression molding, using a weight ratio of 1/25 (sample/KBr). The four-probe method (SP4 probe), with a Lucas/Signatone (California, CA, USA) head with 0.04” of space among the tips of the tungsten carbide material, and a tip radius of 0.0016” coupled to a source meter (Keithley instrument, 2400 series, Cleveland, OH, USA), was used to measure the resistance and to calculate the electrical conductivity of rectangular plate-shaped films, with the following dimensions: 1 cm width, 1 cm length and 0.25 mm thickness. By the photocurrent technique, the resistivity was measured and the electrical conductivity of films was calculated, and the results were corroborated by the four-probe method. A total of 10 V was applied at three consecutive times: (1) a first time of darkness for 60 s, (2) a second time of illumination for 120 s and, finally, (3) a third time of darkness for 60 s. The measurements were taken from a multimeter coupled to a Keithley 619 with a voltage source programmable 230, and illumination was from a 600 W/m^2^ tungsten-halogen lamp.

## 3. Results and Discussion

In this study, we synthesize quasi-spherical particles of chitosan-AuNps in one step; their microstructure was corroborated by DLS and TEM. Therefore, it can be suggested that AuNPs were formed between the chitosan sub-layers. They remained inside or on the surface of the chitosan structure due to the loss of solubility when the chemical interaction took place between NH_3_^+^ groups and tetrachloroauric ions (AuCl_4_^−^) by oxide reduction (Scheme 1). The particle size distributions of chitosan-AuNPs particles synthesized from different concentrations of HAuCl_4_·3H_2_O (0.4, 0.7, 1.0 and 1.3 mM) were measured by DLS (Figure 1). *Dz*, *Dn* and *PDI* were calculated from distribution data of Figure 1 and reported in Table 1. The particle size of chitosan-AuNPs became smaller as the HAuCl_4_·3H_2_O concentration increased; the *Dn* size calculated for the concentration of 0.4 mM was 209 nm and the concentration of 1.3 mM yielded a value of 90 nm. This behavior could be attributed to the concentration of chitosan used, which was maintained constant for all of the different concentrations of HAuCl_4_·3H_2_O. When the HAuCl_4_·3H_2_O concentration was lower, there was a greater amount of amino-reducing groups of chitosan than of the higher concentrations. Consequently, more AuNPs were synthesized with the lowest size, requiring more chitosan chains to cover and stabilize them, increasing the size and distribution of chitosan-AuNPs. Additionally, it is known that chitosan acts as a reducing agent and stabilizer of AuNPs [28], and can cover these by electrostatic interaction from protonated amine groups when the reduction–oxidation reaction is carried out. Such action produces a contraction of the chitosan chain due to the loss of solubility, the latter one allowing the formation of the spherical particles [29].

The morphology of chitosan-AuNPs and self-assembled chitosan-AuNPs/SWCNT-COOH nanostructures was analyzed from photographs obtained by SEM (Figure 2). The characteristic structure of the quasi-spherical shape of chitosan-AuNPs (Figure 2a,b) [28,30] and aligned SWCNT stripes coated with chitosan-AuNPs and embedded in the PVA film were observed. The *Dn* value, calculated from images of at least 200 quasi-spherical chitosan-AuNPs embedded in the PVA films, was 93 nm for the concentration of 0.4 mM (Figure 2a), and 81 nm for 1.3 mM (Figure 2b). Chitosan-AuNPs sizes observed in the film were smaller than characterized by DLS due to the fragmentation caused by sonication during the film formation step. With respect to the chitosan-AuNPs and self-assembled chitosan-AuNPs/SWCNT-COOH individual nanostructures, they were not possible to be identified by TEM, because the chitosan chains were separated from the AuNPs when the electron beam accelerating voltage was applied in order to increase the magnification, and then the temperature increased at the irradiated point and the chitosan chains were separated by the electron beam, which passed through the sample. This behavior is observed when the chitosan glass transition temperature is reached, generating mobility of the chitosan polymer chains [31], deforming the sample and causing the separation of chitosan chains and collapse of AuNPs. Appendix A shows the chitosan segregated from the AuNPs and agglomerated on the copper grid used in TEM. On the other hand, Figure 2c depicts the final nanostructured films of PVA/chitosan-AuNPs/SWCNT-COOH, where well-ordered stripes of SWCNT-COOH ranging from 500 to 1500 nm coated with chitosan-AuNPs nanostructures with *Dn* values between 40 and 50 nm dispersed in PVA matrix due to physical interactions between protonated amine groups of chitosan and carboxylic acid groups of SWCNT may be observed. In addition, the sonication of chitosan-AuNPs/SWCNT-COOH structures during their preparation in suspension provoked the fragmentation and restructuring into smaller sizes of chitosan-AuNPs obtaining values of 45 ± 5.3 nm, which form self-assembled three-dimensional (3D) structures by the interaction of chitosan-AuNPs with the whole SWCNT-COOH structure. Thus, this interaction can increase the electrical properties, such as the electron transfer because both SWCNT and AuNPs possess conductor properties. Figure 2d presents the schematic processes of the film preparation, which were used to obtain the SEM images: (1) in terms of the PVA/chitosan-AuNPs nanostructured films shown in Figure 2b. These films do not reveal electrical behavior: this could be due to the large separation between the particles of chitosan-AuNPs, and (2) the PVA/chitosan-AuNPs/SWCNT-COOH nanostructured films shown in Figure 2c, where the highest interaction between nanostructures is observed.

Figure 3 demonstrates the UV–Vis absorption spectra of chitosan-AuNPs prepared at different concentrations of HAuCl_4_·3H_2_O. The absorption intensity (localized surface plasmon resonances, LSPRs) of the AuNPs increased, and shifted to larger wavelength values as the concentration of HAuCl_4_·3H_2_O increased. At lower concentrations, the maximal wavelength (*λ_max_*) was ca. 526 nm (for 0.4 and 0.7 mM), whereas at higher concentrations, *λ_max_* was ca. 534 nm (for 1.0 and 1.3 mM); the LSPR values were shifted towards the electromagnetic spectrum of the visible region at higher concentrations. Moreover, the results showed that, for the AuNPs at these wavelengths and single LSPR, their structures are spherical [32,33,34]. Further, chitosan acts as a reducing agent of Au^+3^ and a stabilizing agent of AuNPs with the geometry of a quasi-spherical shape. AuNPs have been verified by UV–Vis absorption spectra, SEM images of chitosan-AuNPs and TEM images of AuNPs [35].

The formation of crystalline domains of AuNPs was demonstrated by XRD diffraction patterns. Figure 4 exhibits the diffraction patterns of AuNPs synthesized from different HAuCl_4_·3H_2_O concentrations (0.4, 0.7, 1.0 and 1.3 mM). The peaks observed at 2θ angles of the crystalline domains were at 38.1, 44.6, 64.5 and 77.84°, which correspond to the crystalline planes of Au (111), (200), (220) and (311), respectively. These crystalline planes presented a face-centered cubic arrangement and can be verified in the powder diffraction pattern of Au (JCPDS card: 04-0784). The broad peak observed at about 2θ = 19.3° is attributed to the presence of chitosan molecules. Chitosan has a high percentage of crystalline regions and can be demonstrated by XRD as a narrow peak; however, when chitosan nanoparticles are formed, the crystalline structure can be destroyed to form a more amorphous structure and may be observed as a broad peak [36]. In this instance, we demonstrated by TEM that gold crystals have quasi-spherical structures, where the crystal size increases proportionally with the HAuCl_4_·3H_2_O concentration (Figure 5). The smallest sizes and less dispersity of AuNPs can be observed in Figure 5a, where the structure is more spherical. However, as the size increases, the particles deform slightly and more particles with different sizes are obtained, increasing the dispersion as in the case of Figure 5b–d. The formation of AuNPs may have two growth mechanisms: (a) sequential growth, where in this mechanism the particles’ size increases through the attachment of atoms towards an already formed nucleus; and (b) growth parallel is characterized by clustering [37]. The parallel mechanism can contribute to the augmentation in the polidispersity in the AuNPs formation stage for the concentrations 0.7, 1 and 1.3 mM. The average crystal size (*L*) was determined for the different concentrations from diffractograms (Table 2), by means of the Sherrer equation [25]:(2)$$L=\frac{K\lambda }{\beta Cos\theta }$$
where λ is the wavelength, K is the constant related to the crystal size, known as the shape factor, which commonly has a value of 0.9, β is the line broadening at one half the maximal intensity and θ is the Bragg angle. *L* was calculated from different crystalline planes including (111), (200), (220) and (311). The *D* values of the AuNPs were obtained from nanoparticles of photographs from TEM (see Figure 5) and calculated from the first statistical moment for the different concentrations, and reported in Table 2, with a minimum of a 100-particle count.
(3)$$D=\frac{\sum_{i=1}^{n}D_{i}n_{i}}{\sum_{i=1}^{n}n_{i}}$$
where *n_i_* and *D_i_* are the number and size of particles, respectively.

The stability of self-assembled chitosan-AuNPs/SWCNT-COOH nanostructures was studied by FTIR, whereas we analyzed the physical interaction between the carboxylic groups of SWCNT-COOH and the amine groups of chitosan-AuNPs. The FTIR spectra of (a) chitosan-AuNPs, (b) SWCNT-COOH and (c) the self-assembly of chitosan-AuNPs/SWCNT-COOH are depicted in Figure 6. In the spectra of chitosan-AuNPs (Figure 6a) and the self-assembled chitosan-AuNPs/SWCNT-COOH nanostructures (Figure 6c), there is a broad band due to an overlap at 3410 cm^−1^, provoked by N-H stretching vibrations of the chitosan structure, and 3448 cm^−1^ due to O-H stretching vibrations of the COOH group of SWCNT. For all spectra, medium and sharp bands are observed at 2924 and 2852 cm^−1^, and these bands are attributed to the C-H symmetric and asymmetric stretching vibrations, respectively. The absorption bands located at 1660 cm^−1^, due to the presence of C=O stretching vibrations of amide I at 1592 cm^−1^, are attributed to the N-H bending of the primary amine, and at 1320 cm^−1^, due to the C-N stretching vibrations of amide III, correspond to chitosan; similar results have been reported by several authors [38,39,40]. The characteristic bands of SWCNT functionalized with carboxylic groups were found at 1632 cm^−1^ due to C=O stretching vibrations that indicated the presence of carboxylic groups. Moreover, the bands at 1537 and 1525 cm^−1^ correspond to low-intensity doublets of C=C stretching modes [41] (Figure 6b). In the self-assembled chitosan-AuNPs/SWCNT-COOH nanostructures spectrum (Figure 6c), a new band at 1578 cm^−1^ was observed as a result of the protonation of the amine groups (NH_3_^+^) [42], and this protonation allows the physical interaction between the carboxylic acid of SWCNT and the protonates amine groups of chitosan-AuNPs to form self-assembled chitosan-AuNPs/SWCNT-COOH nanostructures (Scheme 2).

The behavior of the electrical conductivity of PVA/chitosan-AuNPs/SWCNT-COOH nanostructured films prepared at the different concentrations of SWCNT-AuNPs is presented in Figure 7. It was found that the physical interaction between AuNPs at different particle sizes (13–24 nm) and chitosan-AuNPs with the SWCNT-COOH affected the electrical properties of the films. The electrical conductivity was affected by the change of AuNPs at different sizes and for the SWCNT concentration. For instance, electrical conductivity decreased slightly as the concentration of HAuCl_4_·3H_2_O increased at 0.1 wt.% of SWCNT concentration, and at high concentrations of SWCNT-COOH (0.3 and 0.5 wt.%), the electrical conductivity increased at lower concentrations of HAuCl_4_·3H_2_O. In fact, the highest electrical conductivity (3.7 × 10^−5^ σ/cm) was reached for the concentration of 0.5 wt.% of SWCNT and at the lowest concentration of HAuCl_4_·3H_2_O used in the synthesis of chitosan-AuNPs. The increase in the electrical properties is attributed to the decrease in the Au particle size, where the greatest interaction occurs between the smallest size of AuNPs and the highest concentration of SWCNT-COOH, as the highest contact surface between the two is achieved.

The optical absorption spectra of PVA/chitosan-AuNPs/SWCNT-COOH nanostructured films were obtained to estimate the *Eg* values using the Tauc relation [27], from absorption spectra of the PVA/AuNPs-chitosan/SWCNT-COOH nanostructured films of Figure 8. The *Eg* values of each spectrum calculated by using the Kubelka–Munk function (Equation (1)), assuming direct transition (n = 1/2), are shown in Appendix A. In the Kubelka–Munk equation, n = 1/2 was used because the nanostructures and films were prepared at low temperatures. Therefore, there are no thermal effects that influence the *Eg*, and its structure does not change under the temperature conditions studied [25]. The spectra of Figure 8 exhibit absorption bands between 530 and 590 nm due to the PVA films with chitosan-AuNPs/SWCNT-COOH nanostructures. These absorption bands were chosen to calculate the *Eg* values for each ratio of the concentrations of HAuCl_4_·3H_2_O and SWCNT-COOH used. The *Eg* values calculated for the different concentrations of AuNPs and SWCNT-COOH were from 1.8 to 2.25 eV (Figure 9). This shift is caused by the structural arrangement of polymers-nanostructures, electrostatic interaction of chitosan-AuNPs/SWCNT-COOH and individual band gap values of each material. Besides, the structure of SWCNT-COOH has the lowest values ranging from 0.39 to 1.26 eV [43] and the band gap energy changes according to the structure, diameter and doping [44]. The bang gap energy values obtained of chitosan-AuNPs/SWCNT-COOH films are higher than SWCNT-COOH. This increase is caused because chitosan-AuNPs/SWCNT-COOH nanostructures are embedded into the PVA matrix, where the latter can modify the electron trajectory by the system with the overlapping of valence bands [45], giving rise to decrease the exciton binding energy, and hence increasing the energy required by the electron to pass from the valence band to the conduction band. Lioudakis et al. [46] studied the influence of the concentration of SWCNT in a polymer on excitons. They showed that the absorption properties depend on the concentration of the SWCNT and the polymer in which it is embedded. They concluded that the addition of nanotubes embedded into the polymeric matrix shifted the excitonic peaks to larger energies. The SWCNT concentrations in the polymer were from 0.1 to 1 wt.%, with band gap energy values greater than 2 eV for the concentration of 1 wt.%.

Figure 9 reveals *Eg*, determined from the absorption spectra of PVA/AuNPs-chitosan/SWCNT-COOH nanostructured films of Figure 8, as a function of the concentration of HAuCl_4_·3H_2_O used to synthesize chitosan-AuNPs and for the different SWCNT-COOH concentrations utilized to prepare the films. The film that presented the smallest value of *Eg* was for 0.4 mM of HAuCl_4_·3H_2_O and the highest concentration (0.5 wt.%) of SWCNT-COOH. Therefore, there is a greater distribution in all film surfaces, with smaller AuNPs sizes. This contributes to enabling the transport of electrons in the film at higher concentrations of SWCNT-COOH. This behavior can be understood by analyzing the electrical characteristics of each material as follows: (a) on the plasmon resonance surface (PRS) of gold nanoparticles, conductivity of electrons occurs due to the resonant excitation by the incident of photons [47], which increases at smaller particles sizes, and (b) the transport of electrons is higher when there are SWCNT networks rather than individual SWCNTs [48]. Additionally, the film contains GC, which has an important role in decreasing in the *Eg*, because it works as a connector between each space of self-assembled chitosan-AuNPs/SWCNT-COOH nanostructures in order to allow the electrons flow due to its high dielectric constant value (40.1) [49]. Therefore, in maintaining a constant concentration of glycerol and a higher concentration of SWCNT-COOH, there is more interaction between chitosan-AuNPs and SWCNT. The latter gives rise to electron transfer, during which there occurs an increase in electrical properties.

The response of photons to the light exposure of PVA/chitosan-AuNPs/SWCNT-COOH nanostructured films at the different concentrations of HAuCl_4_·3H_2_O and SWCNT-COOH is depicted in Figure 10. The study of the photoresponsive properties that contain different concentrations of self-assembled chitosan-AuNPs/SWCNT-COOH was analyzed in the first 60 s, when the light is turned off, and when the current is low due to the absence of photons and the low density of dislocations and defect points in the material [50]. After this time, when the light is turned on and hits the film, the current increases until it achieves a maximal value for all HAuCl_4_·3H_2_O and SWCNT-COOH concentrations. The highest current values of the photocurrent were obtained for the 0.4 mM concentration of HAuCl_4_·3H_2_O, the values of which were 4.4 × 10^−4^ A (Figure 10a), 6 × 10^−4^ A (Figure 10b) and 6.5 × 10^−4^ A (Figure 10c) for the SWCNT-COOH concentrations of 0.1, 0.3 and 0.5 wt.%, respectively. Moreover, we observed an increase in the current as the crystal size of gold decreased for each SWCNT-COOH concentration. This behavior is common, and is caused by electron transitions and the recombination process, depending on the surface that provides the gold nanoparticles [51]. The electrical conductivity of films was calculated from each maximum value of photocurrent for all HAuCl_4_·3H_2_O and SWCNT-COOH concentrations, which can be observed in Figure 10d. Again, it was demonstrated by photocurrent that the smallest particle size of AuNPs and the highest concentration of SWCNT-COOH (0.4 mM of HAuCl_4_·3H_2_O and 0.5 wt.% of SWCNT-COOH) possess more electrical conductivity in the range of 10^−4^ S/cm. Therefore, smaller-sized AuNPs generate more electrons due to the absorption of photons, while for a higher quantity of SWCNT-COOH, there is more interaction between chitosan-AuNPs and SWCNT-COOH, increasing the electron flow in the film.

## 4. Conclusions

In this work, we prepared novel polymeric films with self-assembled nanostructures of chitosan-AuNPs/SWCNT-COOH. The method used for the preparation consisted of three steps: (a) synthesis of nanoparticles, (b) preparation of self-assembled nanostructures and (c) the embedding of the self-assembled nanostructures into the polymeric film. The films that only had AuNPs or SWCNT-COOH did not present a response to the electrical conductivity. In a particular case, namely the formation of PVA films with embedded SWCNT-COOH, homogeneous dispersion was impossible due to the low solubility of SWCNT-COOH, forming agglomerates of SWCNT-COOH to yield films with zero electrical conductivity. Therefore, the protonation of the amine groups of chitosan induced the electrostatic interaction between chitosan-AuNPs and SWCNT-COOH, which was demonstrated by FTIR, and this process allowed the formation of the nanostructures and their stability into the films. The size control of AuNPs with the increase in the concentration of HAuCl_4_·3H_2_O was demonstrated by TEM and XRD. The homogeneous distribution of self-assembled chitosan-AuNPs/SWCNT-COOH nanostructures can be observed in the SEM photographs. The electrical properties of films increased when the particle size of AuNPs was smaller and the concentration of SWCNT-COOH was higher, and this behavior was caused by the increase in the contact surface area. Furthermore, the films have sensitivity to visible light, increasing the electrical conductivity up to an order of magnitude.

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
