# Peer review of "Characterization and Electrical Properties of PVA Films with Self-Assembled Chitosan-AuNPs/SWCNT-COOH Nanostructures"

_materials, 2020, doi:10.3390/ma13184138_

Round 1

Reviewer 1 Report

The authors present “Characterization and electrical properties of PVA films with self-assembled chitosan-AuNPs/SWCNT-COOH nanostructures”, which systematically described the Materials characteristics, microstructure, optical and electrical conductivity properties. The manuscript itself is interesting, scientific and useful investigation for the biosensors applications. However, before this manuscript is accepted there are a lot of problems need to be addressed:

  1. The English needs to be further polished in the revised manuscript such as: average (line 24);4 mM (line 30);nanostructures (line 39);60 °C (line 39);30 kV, 15 mA(lines 126-127);60 s (line 140);…. etc.
  2. (1) It is suggested that the authors amend the second part of the article to "Results and Discussions". (2) The format of some paragraphs and equations in the text needs to be revised.
  3. It is recommended that the magnification of the SEM images in Figure 2(a), (b), and (c) should be the same.
  4. Transmittance spectra should be added to the text and explain how to obtain different Eg values?
  5. The numbers on the vertical axis in Figure 9, please use scientific notation, i.e., 5.5×10-5.

I recommend that the paper will be published in Materials with minor revision.

Author Response

Characterization and electrical properties of PVA films with self-assembled chitosan-AuNPs/SWCNT-COOH nanostructures

Israel Ceja 4, Karla Josefina González-Íñiguez 5, Alejandra Carreón-Álvarez 6, Gabriel Landazuri 3, Arturo Barrera 2, José Eduardo Casillas 1, Víctor Vladimir A. Fernández 1, and Jacobo Aguilar 1*

Response to Reviewer 1

The authors present “Characterization and electrical properties of PVA films with self-assembled chitosan-AuNPs/SWCNT-COOH nanostructures”, which systematically described the Materials characteristics, microstructure, optical and electrical conductivity properties. The manuscript itself is interesting, scientific and useful investigation for the biosensors applications. However, before this manuscript is accepted there are a lot of problems need to be addressed:

 We thank the reviewer for having examined the manuscript very carefully and for the remarks and suggestions for improving it. We wish to resubmit a modified version of the manuscript, which includes the following suggestions. We have made changes accordingly and given hopefully convincing replies.

Point 1: The English needs to be further polished in the revised manuscript such as: average (line 24);4 mM (line 30); nanostructures (line 39); 60 °C (line 39); 30 kV, 15 mA(lines 126-127); 60 s (line 140); …. etc.

The revised manuscript was carefully checked for spelling and typing errors and corrections were made. We apologize for the mistakes.

Point 2: (1) It is suggested that the authors amend the second part of the article to "Results and Discussions". (2) The format of some paragraphs and equations in the text needs to be revised.

We very much appreciate the reviewer´s opinion. Paragraphs and equations in the revised manuscript were revised and corrections were made.

Point 3: It is recommended that the magnification of the SEM images in Figure 2(a), (b), and (c) should be the same.

SEM images were collocated to the same scale or magnification.

Point 4: Transmittance spectra should be added to the text and explain how to obtain different Eg values?

Transmittance spectra were added to the text as Figure 8. At same time, it was explained the method to obtain the Eg values, and it was redacted the discussion of figure.

Point 5: The numbers on the vertical axis in Figure 9, please use scientific notation, i.e., 5.5×10-5.

The scientific notation was modified as suggested.

Reviewer 2 Report

In this paper, the authors studied in detail the process of formation of chitosan-AuNps quasi-spherical particles. The internal structure of these particles was studied by DLS and TEM methods. The main result of this study is that the particle size of chitosan-AuNPs becomes smaller as the concentration of HAuCl4•3H2O increases. This behavior is explained in this paper by the concentration of chitosan used, which was kept constant for all different concentrations of HAuCl4•3H2O. The paper contains comprehensive experimental data on the analysis of the structure and properties of quasi-spherical particles of chitosan-AuNps and nanostructures of chitosan-AuNPs/SWCNT-COOH. It is shown that PVA films with self-Assembly of chitosan-AuNPS/SWCNT-COOH reveal a photoelectric effect in contrast to nanostructured films with AuNPs only. The authors explain all the observed effects in the article. There are enough figures and photographs in the article, and an comprehensive list of references is provided. I believe that this article can be published in "Materials".

Author Response

Characterization and electrical properties of PVA films with self-assembled chitosan-AuNPs/SWCNT-COOH nanostructures

Israel Ceja 4, Karla Josefina Gónzález-Íñiguez 5, Alejandra Carreón-Álvarez 6, Gabriel Landazuri 3, Arturo Barrera 2, José Eduardo Casillas 1, Víctor Vladimir A. Fernández 1, and Jacobo Aguilar 1*

Reviewer 2

In this paper, the authors studied in detail the process of formation of chitosan-AuNps quasi-spherical particles. The internal structure of these particles was studied by DLS and TEM methods. The main result of this study is that the particle size of chitosan-AuNPs becomes smaller as the concentration of HAuCl4•3H2O increases. This behavior is explained in this paper by the concentration of chitosan used, which was kept constant for all different concentrations of HAuCl4•3H2O. The paper contains comprehensive experimental data on the analysis of the structure and properties of quasi-spherical particles of chitosan-AuNps and nanostructures of chitosan-AuNPs/SWCNT-COOH. It is shown that PVA films with self-Assembly of chitosan-AuNPS/SWCNT-COOH reveal a photoelectric effect in contrast to nanostructured films with AuNPs only. The authors explain all the observed effects in the article. There are enough figures and photographs in the article, and an comprehensive list of references is provided. I believe that this article can be published in "Materials".

We thank the reviewer for having examined the manuscript very carefully and we appreciate the reviewer the comments. At the same time, i share analysis made to this paper: the particle size of AuNPs can be controlled by the concentration of HAuCl4•3H2O, the structure of chitosan has an important role in the preparation of films with self-assembled nanostructures, because amino groups of chitosan induce the interaction with carboxilic acids of single-wall carbon nanotubes, allowing electrostatic interaction between both structures, and finally, the nanostructured materials showed sensitivity to light, increasing electrical conductivity up to an order of magnitude. We again appreciate the carefully review of the manuscript

Reviewer 3 Report

The authors demonstrate the fabrication of PVA films with self-assembled chitosan-AuNPs/SWCNT-COOH nanostructures and characterization of structure and electrical properties. The authors have synthesized the Au nanoparticles using chitosan as a reductant and a stabilizer and which are self-assembled with CNT and PVA. Although the systematic studies are conducted, it is hard to find the novelty of the materials and a new scientific result in this manuscript research and the results are premature for the publication.

Author Response

Characterization and electrical properties of PVA films with self-assembled chitosan-AuNPs/SWCNT-COOH nanostructures

Israel Ceja 4, Karla Josefina Gónzález-Íñiguez 5, Alejandra Carreón-Álvarez 6, Gabriel Landazuri 3, Arturo Barrera 2, José Eduardo Casillas 1, Víctor Vladimir A. Fernández 1, and Jacobo Aguilar 1*

Reviewer 3

The authors demonstrate the fabrication of PVA films with self-assembled chitosan-AuNPs/SWCNT-COOH nanostructures and characterization of structure and electrical properties. The authors have synthesized the Au nanoparticles using chitosan as a reductant and a stabilizer and which are self-assembled with CNT and PVA. Although the systematic studies are conducted, it is hard to find the novelty of the materials and a new scientific result in this manuscript research and the results are premature for the publication.

Response: We appreciate the reviewer´s opinion, however, we should say that we do not agree with him. To our knowledge, and despite the large amount of researches about the synthesis of self-assembled nanostructures in polymer films, very few studies have been reported of ordered arrays of self-assembles nanostructures of chitosan-AuNPs attached to carbon nanotubes into a 2D PVA thin films (as was demonstrated in Figure 2c) with electrical and optical properties (as was showed in Figures 7 and 8, respectively). For instance, in literature there are few published works with the formation of similar structures to those reported. However, those materials were prepared with different methods than those used in this work. The nanomaterials reported by us were fabricated with an easy and novel self-assembly method, that allows the control of the particle size and polydispersity of chitosan-AuNPs with the concentration of HAuCl4·3H2O. Thus, the electrical and optical properties (conductivity and Band-Gap energy) can be controlled by choosing the appropriate concentration of HAuCl4·3H2O. Moreover, the introduction and conclusions in the revised manuscript were re-written, in order to point out clearly the importance and novelty of this work. More recent references were added to the revised manuscript to point out the scientific importance of these materials.

We also disagree that our results are premature for publication. We consider that our experiments are done thoroughly and completely and that we present convincing data relating to the formation of novel nanostructures with highly scientific interest and we performed a complete characterization of their properties. However, to improve the manuscript as was requested by the other reviewers, the experimental methodology and the conclusions were re-written and new explications were added to the results in the revised manuscript.

Reviewer 4 Report

The manuscript presents the study of novel PVA/chitosan-AuNPS/SWCNT-COOH films and their interesting properties. However, some issues must be addressed.
The English language and style throughout the manuscript need to be considerable improved.
Abbreviations should be only used after they were defined first in text (e.g. Dn and Dz – line 24, XRD - line 25).
In the Introduction section, I suggest pointing out what is original about the films proposed and what new findings provide their study compared to other similar studies performed in this field.
Line 76: it is not clear from were the PVA was purchased. Is it Golden bell?
Line 127: 1.5406 Å
Line 150: HAuCl4 is written twice.
For most of the figures, the text on it is too big as compared to the text of the manuscript and must be reduced.
In figure 5, the images are not very different and it must be discussed in more details in the text. What is the conclusion for this figure? Is there any other relevant information that can be given?
In the Conclusion section, emphasis should be made on why is better to combine SWCNT-COOH with AuNPs instead of using PVA films with just SWCNT.

Author Response

Characterization and electrical properties of PVA films with self-assembled chitosan-AuNPs/SWCNT-COOH nanostructures

Israel Ceja 4, Karla Josefina González-Íñiguez 5, Alejandra Carreón-Álvarez 6, Gabriel Landazuri 3, Arturo Barrera 2, José Eduardo Casillas 1, Víctor Vladimir A. Fernández 1, and Jacobo Aguilar 1*

Reviewer 4

We thank the reviewer for having examined the manuscript very carefully and for the remarks and suggestions for improving it. We wish to resubmit a modified version of the manuscript, which includes the following suggestions. We have made changes accordingly and given hopefully convincing replies.

Point 1: The manuscript presents the study of novel PVA/chitosan-AuNPS/SWCNT-COOH films and their interesting properties. However, some issues must be addressed.
The English language and style throughout the manuscript need to be considerable improved.
Abbreviations should be only used after they were defined first in text (e.g. Dn and Dz – line 24, XRD - line 25). 

We very much appreciate the reviewer´s opinion. The correction was made in the revised manuscript, and it was carefully checked for spelling and typing errors. We apologize for the mistakes.

Point 2: In the Introduction section, I suggest pointing out what is original about the films proposed and what new findings provide their study compared to other similar studies performed in this field

A high percentage of the introduction in the revised manuscript was re-written, in order to point out clearly the importance and novelty of this work. More recent references were added to the revised manuscript to point out the scientific importance of these materials. However, in literature there are few published works with the formation of similar structures to those reported

Point 3: Line 76: it is not clear from were the PVA was purchased. Is it Golden bell?

Re: The correction was made in the revised manuscript.

Point 4: Line 127: 1.5406 Å

Re: The correction was made in the revised manuscript.

Point 5: Line 150: HAuCl4 is written twice.

Re: The correction was made in the revised manuscript.

Point 6: For most of the figures, the text on it is too big as compared to the text of the manuscript and must be reduced.

Re: The correction was made in the revised manuscript.

Point 6: In figure 5, the images are not very different and it must be discussed in more details in the text. What is the conclusion for this figure? Is there any other relevant information that can be given?

The discussion and conclusion of the Figure 5 images were written in the manuscript.

Point 7: In the Conclusion section, emphasis should be made on why is better to combine SWCNT-COOH with AuNPs instead of using PVA films with just SWCNT.

The conclusions in the revised manuscript were re-written to emphasize the importance of combining SWCNT-COOH with chitosan-AuNPs, in order to point out clearly the importance and novelty of this work

Round 2

Reviewer 3 Report

  1. Please provide TEM images of chitosan-AuNPs with different sizes and chitosan-AuNPs/SWCNT-COOH to confirmed that the self-assembled structures of Au NPs as the author proposed.
  2. The size of chitosan-AuNPs in Figure 2a and b looks different compared to the size of chitosan-AuNPs in Figure 2c. Please include the size information of chitosan-AuNPs formed with SWCNT-COOH. Why are the two samples different in size?
  3. In figure 8, why there is a valley at 640 nm? Why the intensity on 600nm ~800nm wavelength increases? 
  4. In Figure 9, what material's Eg is this graph? and why Eg is varied depending on concentration of HAuCl4 3H2O salts?

Author Response

Characterization and electrical properties of PVA films with self-assembled chitosan-AuNPs/SWCNT-COOH nanostructures

Israel Ceja 4, Karla Josefina Gónzález-Íñiguez 5, Alejandra Carreón-Álvarez 6, Gabriel Landazuri 3, Arturo Barrera 2, José Eduardo Casillas 1, Víctor Vladimir A. Fernández 1, and Jacobo Aguilar 1* 

Reviewer 3

We thank the reviewer for having examined the manuscript very carefully and for the remarks and suggestions for improving it. We wish to resubmit a modified version of the manuscript, which includes the following suggestions. We have made changes accordingly and given hopefully convincing replies. The revised manuscript was carefully checked for spelling and typing errors and corrections were made. We apologize for the mistakes.

Point 1: Please provide TEM images of chitosan-AuNPs with different sizes and chitosan-AuNPs/SWCNT-COOH to confirmed that the self-assembled structures of Au NPs as the author proposed.

Photographs of chitosan-AuNPs and chitosan-AuNPs/SWCNT-COOH were impossible to identify by TEM. Electron beam accelerating voltage applied to observe these nanostructures is high, increasing the temperature in the sample (chitosan) higher than its glass transition temperature. Under these conditions, the chitosan chains begin to move, separating from the nanoparticles and from SWCNT-COOH, observing collapsed nanostructures. This behavior was explained in the description Figure 2 (from line 263 to 289) of the paper and TEM image was included (Figure S1).

Point 2: The size of chitosan-AuNPs in Figure 2a and b looks different compared to the size of chitosan-AuNPs in Figure 2c. Please include the size information of chitosan-AuNPs formed with SWCNT-COOH. Why are the two samples different in size?

The chitosan-AuNPs structures size in Figure 2a and 2b is different to the Figure 2c, because the chitosan-AuNPs structures were embedded directly in the PVA film to be able to observe these nanostructures. En SEM the voltage applied it is lower; therefore the temperature does not affect the chitosan-AuNPs. The description was collocated in the text from line 262 to 263.

Why are the two samples different in size?

The chitosan-AuNPs/SWCNT-COOH nanostructures were prepared under sonication, in which fragmentation and restructuring occurred before being embedded in the film, decreasing its size. The description was collocated in the text from line 293 to 296.

Point 3: In figure 8, why there is a valley at 640 nm? Why the intensity on 600nm ~800nm wavelength increases? 

why there is a valley at 640 nm?

The valley appears because there is no absorption in this region of UV-Vis spectrum of the structures studied.

Why the intensity on 600nm ~800nm wavelength increases? 

The absorption in this region (at 658) was explained from line 448 to 451.

Point 4: In Figure 9, what material's Eg is this graph? and why Eg is varied depending on concentration of HAuCl4 3H2O salts?

In Figure 9, what material's Eg is this graph?

Is the band gap energy (Eg) of PVA films with self-assembled /AuNPs-chitosan/SWCNT-COOH nanostructures.

Why Eg is varied depending on concentration of HAuCl4 3H2O salts?

The description of behavior in the line 473: “Eg increased with the augmentation in the concentration of HAuCl4·3H2O used (or particle size)” was eliminated, because is a description erroneous that should have been eliminated before. There is no Eg for AuNPs. However, the description about the band gap energy of SWCNT-COOH may change depending on the size of AuNPS due to structural change o conformation of chitosan-AuNPS/SWCNT-COOH, such as was collocated in the text (from line 448 to 451).

Reviewer 4 Report

Thank you for addressing my suggestions. The manuscript was improved as requested.

Author Response

Characterization and electrical properties of PVA films with self-assembled chitosan-AuNPs/SWCNT-COOH nanostructures

Israel Ceja 4, Karla Josefina González-Íñiguez 5, Alejandra Carreón-Álvarez 6, Gabriel Landazuri 3, Arturo Barrera 2, José Eduardo Casillas 1, Víctor Vladimir A. Fernández 1, and Jacobo Aguilar 1*

Reviewer 4

Again, we thank the reviewer for having examined the manuscript very carefully and for the remarks and suggestions for improving it.

Round 3

Reviewer 3 Report

1. the author mentioned "the high interaction with chitosan-AuNPs caused the decrease in the band gap energy". But Eg of your samples is approximately 2 eV (figure 9) therefore it cause increase of Eg. In addition, this value seems too high for Eg of CNT. Why is it too high? Why Au NPs causes sugh a dramatic increase of Eg in CNT? Please describes the literature results reporting CNT which have 2 eV of Eg with citation. 

2. Kubulka plot is not appropriately drawn. It should have high absorption in low wavelength region(high energy) at least you mentioned a semiconductor band gap of CNT. This should be corrected before publication. Mark the line in the KM plot to define the Eg of each sample. 

Author Response

Characterization and electrical properties of PVA films with self-assembled chitosan-AuNPs/SWCNT-COOH nanostructures

Israel Ceja 4, Karla Josefina Gónzález-Íñiguez 5, Alejandra Carreón-Álvarez 6, Gabriel Landazuri 3, Arturo Barrera 2, José Eduardo Casillas 1, Víctor Vladimir A. Fernández 1, and Jacobo Aguilar 1*

Reviewer 3

We thank you the reviewer again for having examined the manuscript very carefully and for the remarks and suggestions for improving it. We apologize for the mistakes.

Point 1: The author mentioned "the high interaction with chitosan-AuNPs caused the decrease in the band gap energy". But Eg of your samples is approximately 2 eV (figure 9) therefore it cause increase of Eg. In addition, this value seems too high for Eg of CNT. Why is it too high? Why Au NPs causes sugh a dramatic increase of Eg in CNT? Please describes the literature results reporting CNT which have 2 eV of Eg with citation. 

Response: Reviewer is right, Eg of chitosan-AuNPs/SWCNT-COOH nanostructures increase with increase HAuCl4·3H2O concentration or increase in the AuNPs size. Therefore, the complete paragraph was rewritten for a better understanding in the lines 373-374 in the revised manuscript.

On the other hand, the higher Eg values for chitosan-AuNPs/SWCNT are due to the electrostatic interactions between chitosan and functionalized SWCNT, which was demonstrated by FTIR, and the SWCNT-COOH embedded in PVA matrix give rise to decrease the exciton binding energies, affecting their optical spectra, and increasing the Eg values. We have added this description on lines 374 to 383 of the revised manuscript in order to clarify this part.

With respect to describing the literature reports of SWCNT with values of Eg of 2 eV, we added a new reference (Lioudakis et al 2008), and the description of this publication was added in lines 383 to 387 in the revised manuscript.

Point 2: Kubulka plot is not appropriately drawn. It should have high absorption in low wavelength region(high energy) at least you mentioned a semiconductor band gap of CNT. This should be corrected before publication. Mark the line in the KM plot to define the Eg of each sample.

Response: We very much appreciate the reviewer´s opinion. We checked the data Figure 8 in order to be sure that it was correct. Figure 8 is it graphed the diffuse reflectance UV-Vis spectra of film and was obtained by plotting F(R) (Kubelka-Munk function) vs wavelength.

Then, with regard to the high absorption in low wavelength region (high energy), this behavior does not occur in the spectra of Figure 8, because the SWCNT concentrations are low, and most of SWCNT are covered by chitosan-AuNPs nanostructures, where these do not absorb at the same wavelength of the SWCNT. In Figure 8, there is an absorption region (approximately between 200 y 350 nm), noise is observed that does not allow Eg to be calculated. The calculated semiconductor band gap of CNT was obtained from absorption region between 638 and 700 nm, with average absorption edge of 658 (1.5 eV), which correspond to absorption band of CNT semiconductors. Spectra of our system are very different from those of substances with high concentration reported elsewhere.

Thus, systems with high absorption in low wavelength region (high energy) occur when composition of the CNT to be studied is high or it is a single sample only, or the components of the sample with CNT absorb at the same wavelength.

As the reviewer requested, we added the KM plots (known as Tauc´s graph ((F(R)hv)2 of Kubelka-Munk function versus hv), in the supplementary materials as Figures S2-S6, and Eg is shown with a red line (extrapolation of the straight line of the curve to zero in the hv axis) for the different concentrations of AuNPs and SWCNT-COOH used. We do not put the figures in the main text, because there is too many graphs for the calculation of Eg and it can generate confusion.